# Who would take part in a pandemic preparedness cohort study? The role of vaccine-related affective polarisation: Cross-sectional survey

Aziz Mert Ipekci[1,2,3], Eva Maria Hodel[1,3], Maximilian Filsinger[3,4,5], Selina Wegmüller[3,6,7], Simone Schuller[3,8], Markus Freitag[3,5], Annika Frahsa[1,3], Gilles Wandeler[3,9], Nicola Low[1,3]*

1 Institute of Social and Preventive Medicine, University of Bern, Bern, Switzerland, 2 Graduate School for Health Sciences, University of Bern, Bern, Switzerland, 3 Multidisciplinary Center for Infectious Diseases, University of Bern, Bern, Switzerland, 4 ESPOL-LAB, Université Catholique de Lille, Lille, France, 5 Institute of Political Science, University of Bern, Bern, Switzerland Markus Freitag, 6 Department of Clinical Research, CTU Bern, University of Bern, Bern, Switzerland, 7 Institute of Philosophy, University of Bern, Bern, Switzerland, 8 Department of Clinical Veterinary Medicine, University of Bern, Switzerland, 9 Department of Infectious Diseases, Bern University Hospital, University of Bern, Bern, Switzerland

* nicola.low@unibe.ch

## Abstract

Cohort studies are an important tool for public health research about emerging infectious diseases. Willingness to participate in research is associated with multiple factors. Little is known about the role of people's feelings, especially in the aftermath of the COVID-19 pandemic. Affective polarisation, an affinity for people with similar attitudes to oneself and hostility toward those with opposing views, is a measure of feelings about issues, including COVID-19 vaccination. The aim of this study was to investigate factors associated with willingness to participate in a future household-based cohort study on pandemic preparedness. We did a cross-sectional online survey in the Canton of Bern, Switzerland. We invited a random sample of persons aged 18 + years from 15,000 private households. We asked about willingness to take part and collected data about demographic, social and household characteristics, and affective polarisation related to COVID-19 vaccination. We performed univariable and multivariable analyses, weighted by household size to estimate odds ratios (with 95% confidence intervals, CI). Among responders, 49.8% (95% CI 47.9–51.8, weighted proportion) were willing to take part in a cohort study. In multivariable analysis, higher educational level (adjusted odds ratio 2.48, 95% CI 1.81–3.39) and higher monthly income (1.92, 1.39–2.65) were most strongly associated with higher willingness to participate. Opposition to COVID-19 vaccination was associated with lower willingness to participate (0.53, 0.39–0.72). Affective polarisation modified the relationship between vaccination attitudes and willingness to participate. Compared with non-polarised vaccination supporters, polarised supporters were more willing to participate (marginal adjusted odds ratio 1.51, 1.05–2.16), whereas willingness to participate was lower among both non-polarised (0.53, 0.32–0.86) and affectively

**Data availability statement:** The data from this study are not publicly available because of the risk of identification of an individual participant. Some municipalities in the study area had very few respondents, and the characteristics reported could be combined to identify a person. Data are available upon reasonable request. Researchers interested in accessing the data from this online survey can request it by contacting the project management team at BEready.mcid@unibe.ch or by filling in the request form (https://www.beready.unibe.ch/for_researchers/). De-identified data will be made available to qualified researchers for approved analyses, and access will be granted following review and approval of a study proposal, establishment of a data use and transfer agreement, and successful ethical approval. Restrictions may apply depending on the specific nature of the data requested to ensure participant privacy and compliance with institutional policies. A shared dataset will exclude study participants from municipalities with fewer than 5 respondents. Fees may apply for data preparation and transfer, depending on the nature of the collaboration. We encourage multidisciplinary collaboration on projects that align with the goals of our study, particularly those focused on pandemic preparedness.

**Funding:** This study was financially supported by the Multidisciplinary Center for Infectious Diseases at the University of Bern in the form of a project grant awarded to AF, MF, and NL (MA21). This study received additional funding from the Multidisciplinary Center for Infectious Diseases at the University of Bern in the form of a grant to the BEready project awarded to NL and GW. No additional external funding was received for this study. The funders had no role in study design, data collection and analysis, decision to publish, or preparation of the manuscript.

**Competing interests:** The authors have read the journal's policy and have the following competing interests: AF is a representative for the German Speaking Network of Participatory Health Research. This does not alter our adherence to PLOS ONE policies on sharing data and materials.

polarised (0.26, 0.12–0.56) vaccination opposers. Willingness to participate in a cohort study was moderate. Following the COVID-19 pandemic, addressing affective polarisation, as well as socioeconomic factors, is needed to improve participation in pandemic preparedness research.

## Introduction

Cohort studies are an important tool for public health research about emerging infectious diseases and pandemics [1]. At the start of the COVID-19 pandemic in early 2020, policy makers needed to know the proportion of the population who had already been exposed to severe acute respiratory syndrome coronavirus 2 (SARS-CoV-2) to plan appropriate prevention measures. Researchers in Switzerland were among the first to provide this information, by contacting participants in an existing cohort study on non-communicable diseases and obtaining blood samples for SARS-CoV-2 antibody testing [2]. Seroprevalence was 10%, showing that most people were still susceptible to SARS-CoV-2, guiding the need for ongoing measures to prevent transmission in the population with the minimum of restrictions [2]. In anticipation of the next pandemic, global health experts have proposed that 'pandemic pre-paredness cohorts' can establish the research infrastructure and then pivot quickly to collect relevant data when a new pandemic pathogen starts to spread [1]. New cohort studies in this field are needed because existing studies, established for other reasons, have not necessarily collected the data needed to improve understanding about behaviours and exposures associated with susceptibility, transmission and immunity of emerging pandemic pathogens.

People's feelings and concerns about public health issues may influence their willingness to participate in research in a new field and should be considered when designing and starting studies [3]. Documented reasons for longstanding declines in participation in population-based research include limited time, declines in altruism and community involvement in general, and mistrust in science [4]. Previous studies have focused on socioeconomic and demographic factors associated with willingness to participate in health research, and on views about issues such as data privacy [4–6]. The novelty and scale of the COVID-19 pandemic may now have affected attitudes towards research because of the unprecedented amounts of new research findings [7], awareness about research [8], and uncertainties about communication, and public trust between scientists, policymakers and the public [9–11]. Political polarisation, according to ideology, partisan affiliation or trust in political institutions, has characterised responses to many aspects of COVID-19 prevention [12,13]. The concept of affective polarisation, defined as an affinity for people with similar attitudes to oneself and hostility toward those with opposing views, has been used as a quantitative method to capture peoples' emotions about the opinions of others [14]. Whilst affective polarisation was first defined and studied in the United States of America to understand political opinions [14], the COVID-19 pandemic revealed associations between affective polarisation and health behaviours such as COVID-19 vaccination

uptake, adherence to pandemic countermeasures or anxiety about COVID-19 [14–18]. Measures of people's feelings about public health measures such as COVID-19 vaccination, and how they view others, are more proximate to health-associated behaviours than partisan or ideological differences. Whether affective polarisation is associated with willingness to take part in research about pandemics more generally is not well understood. The objectives of this study were: 1) to investigate sociodemographic factors associated with willingness to participate in a planned pandemic preparedness cohort study; 2) to explore associations between COVID-19 vaccination-related affective polarisation and willingness to participate; and 3) to describe reasons for taking part, or not taking part, in a pandemic preparedness cohort study.

## Materials and methods

We did a cross-sectional survey in the canton of Bern, Switzerland. Bern, population 1,070,898, is the largest of the 26 Swiss cantons geographically, with demographic characteristics similar to those of Switzerland as a whole. The survey was part of the preparation for a population-based pandemic preparedness cohort study, BEready ("Bern, get ready"). The BEready cohort will study whole households [19], including pets [20], because of the importance of within household transmission of infectious agents [21] and a One Health approach to understand and prepare for pandemics which may spread between humans and animals [22]. The Cantonal Ethics Committee in Bern provided an exemption from ethical approval under the Human Research Act because the survey did not ask for any personal health-related information (BASEC-Nr: Req-2022–00877). We report this study following the Strengthening the Reporting of Observational Studies in Epidemiology guideline for cross-sectional studies (research checklist) (S1 Table).

### Study sample

We contacted the Cantonal Administration and Information Office to obtain a random sample of private household addresses in the canton of Bern. The requested sample of 15,000 households was the same as that of a previous online study about willingness to participate in personalised health research in Switzerland [5]. We requested a random sample of 3,000 households, each with 1, 2, 3, 4, or 5 + members, oversampling larger households because their participation is important for a cohort study that aims to investigate household infection transmission. One person aged 18 or older from each household was randomly chosen as the primary contact. The selected household members received a letter by surface mail, which introduced the issue of pandemic preparedness and about potential participation in research about existing and future infectious diseases. The letter contained a personal access code and a link to an online questionnaire. On request, they could receive a printed version of the survey. They were informed that by filling in the questionnaire, they consented to participate in the study. Invitation letters were sent on 19th September 2022, with two reminders on 14th October and 9th November 2022. The survey closed on 18th December 2022.

### Survey development

The survey questions were taken from published national and international questionnaires, whenever possible, including the Swiss Health Study pilot [23], a nationwide survey about personalised health research [24], the COVID-19 social mixing survey [25], the International Social Survey Programme [26] and a study of COVID-19-associated affective polarisation [15]. The survey was available in German, French, Italian, and English. A convenience sample of 10 people speaking the languages of the survey tested the questions and we made changes to improve readability where necessary.

The primary outcome was self-reported willingness to participate in a long-term cohort study about infectious diseases, measured on a 4-point scale, "yes", "probably yes", "probably no", or "no". The survey questions covered: demographic characteristics including age and gender identity, social, and household characteristics, patterns of social mixing and a pre-defined list of motivations to participate or not participate in cohort studies (S2 Table). Opinions about COVID-19 vaccination were measured on a scale from zero (complete rejection) to 10 (complete support). We measured affective polarisation using a feeling thermometer [27]. First, we asked participants about their feelings towards people who get

vaccinated from –5 ("very cold") to +5 ("very warm"). Then, we asked the same question about people who do not get vaccinated. The absolute difference between their answers to the two questions gave the affective polarisation level on a scale from zero to ten [15].

## Patient and public involvement

We did not involve patients or the public in the development of the questionnaire used in this study. After the survey was completed, we put together a community advisory group of representatives and members of the public, who reviewed the study findings. Their feedback is informing community engagement strategies to optimise enrolment into the BEready cohort study.

## Statistical analysis

We used RStudio for all analyses [28]. To account for the oversampling of larger households, we performed inverse probability weighting (S3 Table) using the survey package (version 4.4–2) [29]. We defined respondents as those who answered at least half of the survey questions [5] and included respondents who answered the question about the primary outcome in statistical analyses. We compared responders and non-responders, based on the two variables available, household size and main language spoken at home. We used the municipality of residence to map the proportions of responders willing to participate. We dichotomised three variables (S2 Table, S1 Fig): 1) the primary outcome variable, willingness to participate (yes/no), combining responses "yes" and "probably yes", and "no" and "probably no"; 2) opinion about vaccination (opposition, 0–4, support, 6–10). A response of 5 was considered as neutral and was treated as missing in the main analysis [15]; affective polarisation (not polarised, 0–4, polarised, 5–10). We grouped variables such as education and income into a smaller number of categories, following conventions used in previous publications (S2 Table) [25].

We described our study population characteristics using means (and standard deviation, SD) or median (and interquartile range, IQR) for continuous variables and frequencies and percentages for categorical variables. We used logistic regression to estimate univariable odds ratios (OR and 95% confidence intervals, CI) for the association between reported willingness to participate and exposure variables. We built a multivariable regression model, based on knowledge available on the topic and findings from the univariable analyses. The model included age, gender, education, current work situation, nationality, household income, household size, location, language, opinion about COVID-19 vaccination, and vaccination-related polarisation. We compared models using likelihood ratio tests and expressed associations as adjusted odds ratios, aOR, with 95% CI. To explore affective polarisation related to opinions about COVID-19 vaccination further, we examined the observed association between respondents' support or opposition to COVID-19 vaccination and willingness to participate, according to the presence or absence of affective polarisation. We included an interaction term in the multivariable model and reported marginal odds ratios [30]. Lastly, we summarised motivations for participation or non-participation in a cohort study. Participants could choose multiple reasons.

We conducted three sensitivity analyses: 1) we reclassified people with a score of 5 in the vaccination opinion scale with either supporters of vaccination or as opposing vaccination; 2) we tested alternative cut-off points for affective polarisation (not polarised, 0–3; polarised, 4–6; heavily polarised, 7–10); 3) we varied the definition of a responder by adjusting the minimum number of answered survey questions required for inclusion (33% and 66% considered as responders).

## Results

### Characteristics of the study population

The 15,000 selected households came from 306/335 municipalities in the Canton of Bern. Overall, 3,425/15,000 (22.8%) people from 273 municipalities responded to at least half of the survey and 3,394 (22.6%) responders replied to the question about willingness to participate in a cohort study (S2 Fig). The highest proportion of responders were from 2-person

households (29.2%, 877/3,000), with the lowest proportion from households with 5 or more members (11.7%, 351/3,000) (S4 Table). The distribution of respondents, according to the main language spoken in the household, was similar to the numbers invited (S4 Table). Responses were similar among German-speaking (22.8%, 3,074/13,474) and French-speaking households (21.2%, 312/1,473). Among the remaining households, there were 18.2% (2/11) Italian-speaking and 14.6% (6/41) English-speaking respondents. The percentage of missing values was 0.5% to 4.1% for sociodemographic variables. For the questions related to COVID-19 vaccine, 8.5% and 11.8% did not respond. We did not conduct imputation for missing values.

We present results from the weighted analysis; descriptive results in the unweighted analysis were very similar (S5 Table). Table 1 presents the weighted distribution of the participants' characteristics. Amongst responders, there were more women (256,277, 52.3%) than men (229,354, 46.8%), more than a third (182,256, 37.2%) had tertiary education and two-fifths had a household income between 4,500–9,000 Swiss Francs (205,540, 42.0%).

### Sociodemographic factors and willingness to participate in a long-term cohort study

Overall, 1,660/3,394 (unweighted proportion 48.9%, weighted proportion 49.8%, 95% CI 47.9–51.8) of responders reported being willing to participate in a long-term study. Willingness to participate was higher near the city of Bern than in rural areas (Table 1, Fig 1). The highest levels of willingness were among individuals with tertiary education (weighted proportion 62.6%) and those earning more than 9,000 Swiss Francs per month (64.9%). The lowest levels were seen among individuals with less than compulsory education (31.1%) and those categorised as being opposed to COVID-19 vaccination (32.6%). Similar proportions of women and men were willing to participate. We asked respondents whether they would be willing for their children and pets to participate in a long-term study (Table 1). Among respondents with children, the willingness for their children to participate was lower (40.0%) than for themselves (49.8%). Among those with pets, willingness to include pets varied by species, with the highest participation rates for rodents (58.6%), followed by dogs (47.9%), cats (46.0%), and rabbits (37.9%).

After adjustment for all variables included in the model, willingness to participate was more likely among participants with the highest educational level (aOR 2.48, 95% CI 1.81–3.39) and those with the highest income (aOR 1.92, 1.39–2.65); and was less likely among older participants (aOR 0.99, 0.98–0.99 for a one-year increase) and larger households (5 or more people vs. 1 person, aOR 0.67, 0.44–1.01) (Table 2).

### COVID-19 vaccination-related factors and willingness to participate

Most respondents expressed opinions in support of COVID-19 vaccination (65.8%) (Table 1, S1 Fig). In both univariable and multivariable analyses, people who had negative opinions about COVID-19 vaccination were less willing than those who expressed support for vaccination to say they would participate in a long-term cohort study (aOR 0.53, 0.39–0.72). Amongst all respondents, people with polarised feelings towards others about COVID-19 vaccination were more willing than those defined as non-polarised to participate (aOR 1.51, 1.20–1.89) (Table 2). The direction of the association between COVID-19 vaccination opinions and willingness to participate differed between those who showed affective polarisation and those who were non-polarised (Table 3). The increase in willingness to participate among people with polarised opinions was only observed among those who supported COVID-19 vaccination (aOR 1.51, 1.05–2.16). People with opinions opposed to COVID-19 vaccination were less willing to participate than non-polarised supporters of vaccination, particularly those who were affectively polarised (aOR 0.26, 0.12–0.56). The three sensitivity analyses did not change the results (S6 Table).

### Reasons for participation or non-participation in research

People who answered "yes", "probably yes", and "probably no" to the willingness to participate question were asked about their reasons to participate in a long-term study (n = 2,776). People who answered "no" were asked for reasons to decline

**Table 1. Self-reported willingness to participate in a long-term study about infectious diseases, weighted denominator.**

| | | Denominator N[1] | | Willingness to participate in cohort study, % (95% confidence intervals) |
|---|---|---|---|---|
| **Number of people** | | 489,973 | (100%) | 49.8 (47.9 - 51.8) |
| **Age group, years** | 18-29 | 44,522 | (9.1%) | 58.5 (52.9 - 64.0) |
| | 30-64 | 291,949 | (59.6%) | 52.5 (50.0 - 54.9) |
| | 64+ | 153,501 | (31.3%) | 42.2 (38.5 - 46.0) |
| **Gender** | Female | 256,277 | (52.3%) | 50.5 (47.8 - 53.2) |
| | Male | 229,354 | (46.8%) | 49.2 (46.4 - 52.1) |
| | Other | 1,854 | (0.4%) | 58.7 (27.2 - 85.8) |
| | No response | 2,488 | (0.5%) | 25.2 (5.1 - 58.4) |
| **Education level** | Compulsory education or less | 69,135 | (14.1%) | 31.1 (26.5 - 36.0) |
| | Upper secondary | 225,091 | (46.0%) | 46.0 (43.1 - 48.9) |
| | Tertiary | 182,256 | (37.2%) | 62.6 (59.5 - 65.6) |
| | Other | 4,752 | (1.0%) | 43.9 (23.1 - 66.4) |
| | No response | 8,740 | (1.8%) | 33.2 (20.3 - 48.2) |
| **Current work situation** | Full-time employee | 211,250 | (43.1%) | 54.9 (51.9 - 57.9) |
| | Part-time employee | 97,113 | (19.8%) | 55.5 (51.5 - 59.5) |
| | Not employed | 151,874 | (31.0%) | 41.0 (37.3 - 44.7) |
| | In education | 10,546 | (2.2%) | 60.7 (51.3 - 69.6) |
| | Other | 6,523 | (1.3%) | 32.1 (18.7 - 48.0) |
| | No response | 12,666 | (2.6%) | 27.4 (17.3 - 39.3) |
| **Income, Swiss Francs** | <4,500 | 96,560 | (19.7%) | 43.6 (38.7 - 48.5) |
| | 4,500–9,000 | 205,540 | (42.0%) | 51.6 (48.5 - 54.6) |
| | >9,000 | 105,130 | (21.5%) | 64.9 (61.3 - 68.5) |
| | Other | 62,679 | (12.8%) | 32.7 (27.9 - 37.7) |
| | No response | 20,064 | (4.1%) | 26.9 (19.0 - 35.8) |
| **Household residents, number** | 1 | 185,820 | (38.0%) | 52.6 (48.7 - 56.4) |
| | 2 | 170,629 | (34.8%) | 48.2 (45.0 - 51.5) |
| | 3 | 54,814 | (11.2%) | 51.1 (47.6 - 54.7) |
| | 4 | 53,981 | (11.0%) | 48.5 (45.1 - 51.8) |
| | 5+ | 24,729 | (5.0%) | 40.5 (35.7 - 45.3) |
| **Household location[2]** | Urban | 278,236 | (56.9%) | 52.3 (49.6 - 55.0) |
| | Intermediate | 118,778 | (24.5%) | 48.4 (44.6 - 52.3) |
| | Rural | 90,725 | (18.5%) | 44.3 (40.0 - 48.7) |
| **Nationality** | Swiss | 415,354 | (84.8%) | 50.8 (48.6 - 52.9) |
| | Foreign | 69,632 | (14.2%) | 45.7 (41.1 - 50.3) |
| | No response | 4,987 | (1.0%) | 28.0 (13.0 - 47.2) |
| **Language spoken at home** | German | 428,396 | (87.4%) | 51.1 (48.9 - 53.3) |
| | French | 45,301 | (9.2%) | 49.6 (43.2 - 55.9) |
| | Italian | 6,604 | (1.4%) | 49.4 (32.4 - 66.5) |
| | English | 9,671 | (2.0%) | 54.2 (42.5 - 65.6) |
| **Opinion about vaccination[3]** | For vaccination | 322,536 | (65.8%) | 57.7 (55.3 - 60.1) |
| | Oppose vaccination | 83,395 | (17.0%) | 32.6 (28.4 - 36.9) |
| | No response | 41,576 | (8.5%) | 33.3 (27.3 - 39.7) |

*(Continued)*

**Table 1.** (Continued)

| | | Denominator N[1] | | Willingness to participate in cohort study, % (95% confidence intervals) |
|---|---|---|---|---|
| **Vaccination related affective polarisation** | Polarised | 244,371 | (49.9%) | 56.8 (54.0 - 59.5) |
| | Not polarised | 187,844 | (38.3%) | 44.6 (41.5 - 47.6) |
| | No response | 57,758 | (11.8%) | 37.6 (32.3 - 43.1) |
| | **Willingness to participate with children** | | | |
| | | **Number with children** | | **Willingness to participate** |
| **On behalf of their children** | – | 80,605 | (100%) | 40.0 (37.0 - 42.9) |
| | **Willingness to participate with pets** | | | |
| | | **Number with pets** | | **Willingness to participate** |
| **On behalf of their pets** | Any pet | 134,046 | (100%) | 46.1 (42.6 - 49.6) |
| **Type of pet[4]** | Dogs | 43,521 | (32.5%) | 47.9 (41.8 - 54.1) |
| | Cats | 90,996 | (67.9%) | 46.0 (41.7 - 50.3) |
| | Rabbits | 7,066 | (5.3%) | 37.9 (26.9 - 49.9) |
| | Rodents | 6,686 | (5.0%) | 58.6 (45.2 - 71.2) |
| | Others | 13,575 | (10.1%) | 47.8 (37.6 - 58.0) |

[1]Weighted number of participants

[2]The sum of percentages for the household location variable does not equal 100% due to missing data that were not classified by the Cantonal Administration and Information Office.

[3]The sum of percentages for the opinions on vaccination does not total 100% because responses marked as "5", representing a neutral position at the midpoint of the scale, were excluded from the analysis.

[4]The sum of the percentages exceeds 100% because some households reported owning multiple types of pets.

participation (n = 637) (S7 Table). The most common reasons for participation were: "I can contribute to the health of fellow human beings" (n = 1,559, 56%); "I can contribute to better preparation for the next pandemic" (n = 1,134, 41%), followed by answers such as, "I am interested in research and health" (n = 1,031, 37%), "I will get a free health check" (n = 713, 26%) and "I will receive the study results as feedback" (n = 682, 25%). The main reasons to decline were: lack of interest (n = 266, 42%); privacy concerns and mistrust (Fig 2, S7 Table).

## Discussion

In this cross-sectional online survey, about half of respondents reported willingness to take part in a long-term cohort study for pandemic preparedness, with slightly lower proportions willing for their children or pets to take part. Older adults, individuals with lower levels of education and income, and those opposing COVID-19 vaccination were less likely to report that they would participate. Affective polarisation was associated with increased willingness to participate among those supporting COVID-19 vaccination, whilst people expressing opposition to vaccination were less willing to participate, particularly those who were affectively polarised. The main reasons given for participation in research were to help others and to improve pandemic preparedness, while lack of interest, privacy concerns and mistrust were the most commonly stated reasons for non-participation.

### Strengths and limitations

The design of our study has three main strengths. First, the sample was drawn at random from a population register, with oversampling from larger households, which account for a smaller proportion of households in the canton of Bern. Second, we asked about affective polarisation related to feelings about COVID-19 vaccination, in addition to more traditional

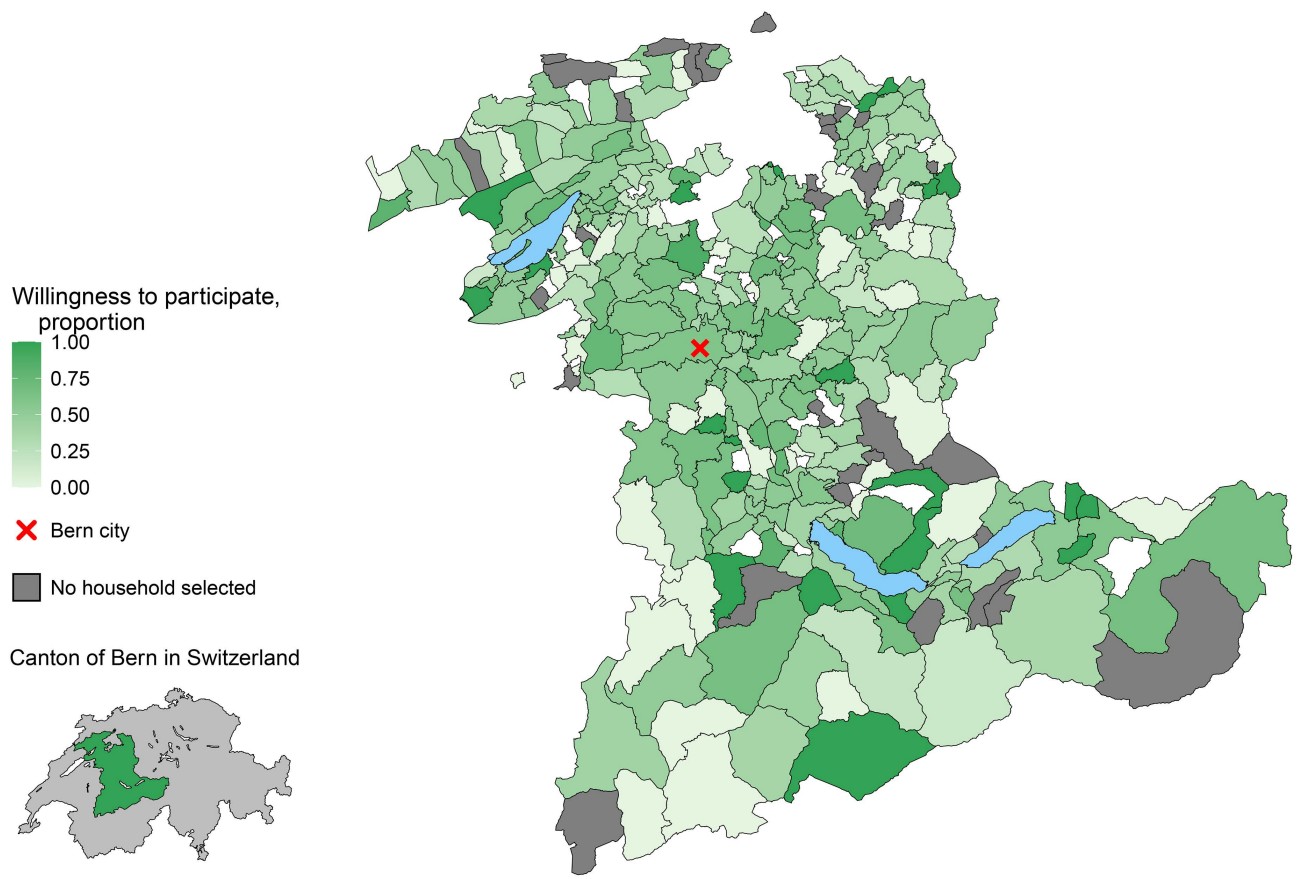

**Fig 1. Proportion of respondents willing to participate, by municipality in the canton of Bern, Switzerland.** The map uses: the swissBOUNDARIES3D dataset and the official directory of towns and cities both published by the Federal Office of Topography swisstopo (swisstopo.admin.ch); and the lakes and rivers 2025 GEOM TK dataset of Swiss Federal Statistical Office (bfs.admin.ch) [31,32].

sociodemographic factors. Third, we used questions that had been used and/or validated in other surveys to facilitate comparisons with other studies. There are three main limitations to the study design. First, the study participants came from a single canton in Switzerland because this is the setting for the planned BEready cohort study. Whilst this may limit generalisability of the study findings, the demographic characteristics of the canton of Bern are similar to those of Switzerland as a whole. Second, for reasons of efficiency, the questionnaire was primarily designed to be answered online and was available in only four languages. The survey was, therefore, not accessible to people without ready access to a computer and the internet, or to those who could not understand one of the survey languages; only 184 people asked for a paper version of the questionnaire. Our findings may therefore be biased if factors limiting participation to those with digital access are also associated with willingness to participate in research. Third, the number of questions in the survey was restricted since a lengthy questionnaire might have decreased the response rate [33]. Whilst there is evidence that political ideology and trust in government institutions are associated with an individual's willingness to participate in scientific research [34,35], we were not able to include questions about a broader range of political variables.

## Comparison with other studies

Assessments of the level of willingness to participate in public health research need to take into account the proportion responding to the online survey. When compared with other online surveys using random sampling approaches, our

**Table 2. Factors associated with willingness to participate, univariable and multivariable logistic regression.**

| Characteristic | Univariable model | | | Multivariable model[1] | | |
|---|---|---|---|---|---|---|
| | OR[2] | (95% CI)[2] | p-value | aOR[3] | (95% CI)[2] | p-value |
| **Age, per year** | 0.99 | (0.98 - 0.99) | <0.001 | 0.99 | (0.98 - 0.99) | <0.001 |
| **Gender** | | | 0.346 | | | 0.165 |
| Female | 1 | | | 1 | | |
| Male | 0.95 | (0.81 - 1.11) | | 0.9 | (0.74 - 1.09) | |
| Other | 1.39 | (0.39 - 4.93) | | 3.13 | (0.87 - 11.3) | |
| No response | 0.33 | (0.09 - 1.24) | | 0.46 | (0.06 - 3.33) | |
| **Education level** | | | <0.001 | | | <0.001 |
| Compulsory education or less | 1 | | | 1 | | |
| Upper secondary education | 1.89 | (1.47 - 2.43) | | 1.59 | (1.19 - 2.12) | |
| Tertiary education | 3.70 | (2.86 - 4.79) | | 2.48 | (1.81 - 3.39) | |
| Other | 1.74 | (0.72 - 4.19) | | 1.57 | (0.49 - 5.05) | |
| No response | 1.10 | (0.57 - 2.12) | | 1.61 | (0.75 - 3.43) | |
| **Current work situation** | | | <0.001 | | | 0.053 |
| Full-time employee | 1 | | | 1 | | |
| Part-time employee | 1.03 | (0.84 - 1.25) | | 1.28 | (0.98 - 1.67) | |
| Not employed | 0.57 | (0.47 - 0.69) | | 0.93 | (0.70 - 1.23) | |
| In education | 1.27 | (0.85 - 1.91) | | 1.91 | (1.15 - 3.15) | |
| Other | 0.39 | (0.20 - 0.77) | | 0.85 | (0.40 - 0.79) | |
| No response | 0.31 | (0.18 - 0.54) | | 0.92 | (0.49 - 1.76) | |
| **Household income, Swiss Francs** | | | <0.001 | | | <0.001 |
| < 4'500 | 1 | | | 1 | | |
| 4'500 − 9'000 | 1.39 | (1.11 - 1.75) | | 1.21 | (0.93 - 1.59) | |
| > 9'000 | 2.40 | (1.87 - 3.08) | | 1.92 | (1.39 - 2.65) | |
| Other | 0.62 | (0.46 - 0.83) | | 0.62 | (0.44 - 0.87) | |
| No response | 0.47 | (0.29 - 0.75) | | 0.46 | (0.26 - 0.83) | |
| **Number of household members** | | | 0.002 | | | <0.001 |
| 1 | 1 | | | 1 | | |
| 2 | 0.84 | (0.69 - 1.03) | | 0.74 | (0.59 - 0.95) | |
| 3 | 0.94 | (0.76 - 1.16) | | 0.65 | (0.50 - 0.85) | |
| 4 | 0.85 | (0.69 - 1.04) | | 0.50 | (0.38 - 0.67) | |
| 5≤ | 0.61 | (0.48 - 0.79) | | 0.67 | (0.44 - 1.01) | |
| **Household location** | | | 0.008 | | | 0.601 |
| Urban | 1 | 1 | | 1 | 1 | |
| Intermediate | 0.86 | (0.71 - 1.03) | | 1.03 | (0.83 - 1.27) | |
| Rural | 0.73 | (0.59 - 0.89) | | 0.90 | (0.71 - 1.14) | |
| **Nationality** | | | 0.013 | | | 0.791 |
| Swiss | 1 | 1 | | 1 | 1 | |
| Foreign | 0.81 | (0.66 - 1.00) | | 0.90 | (0.65 - 1.25) | |
| No response | 0.38 | (0.16 - 0.87) | | 0.84 | (0.30 - 2.31) | |
| **Language** | | | 0.907 | | | 0.729 |
| German | 1 | 1 | | 1 | 1 | |
| French | 0.99 | (0.76 - 1.29) | | 1.11 | (0.80 - 1.54) | |
| Italian | 0.99 | (0.51 - 1.92) | | 1.34 | (0.70 - 2.53) | |
| English | 1.19 | (0.74 - 1.92) | | 0.90 | (0.48 - 1.67) | |
| **Opinion about COVID-19 vaccination** | | | <0.001 | | | <0.001 |

*(Continued)*

**Table 2.** (Continued)

| Characteristic | Univariable model | | | Multivariable model[1] | | |
|---|---|---|---|---|---|---|
| | OR[2] | (95% CI)[2] | p-value | aOR[3] | (95% CI)[2] | p-value |
| Support vaccination | 1 | 1 | | 1 | 1 | |
| Oppose vaccination | 0.35 | (0.28 - 0.44) | | 0.53 | (0.39 - 0.72) | |
| No response | 0.37 | (0.27 - 0.49) | | 0.34 | (0.07 - 1.69) | |
| **Vaccination-related polarisation** | | | <0.001 | | | <0.001 |
| Not polarised | 1 | 1 | | 1 | 1 | |
| Polarised | 1.64 | (1.38 - 1.93) | | 1.51 | (1.20 - 1.89) | |
| No response | 0.75 | (0.58 - 0.97) | | 0.88 | (0.47 - 1.67) | |

[1]Model includes the following variables: Age, gender, education level, current work situation, income, household size, household location, language, nationality, opposition to vaccination, affective polarisation: vaccination opposer (interaction term) and willingness to participate (outcome).

[2]OR, odds ratio, CI, confidence interval.

[3]aOR, adjusted odds ratio.

study's response rate (23%) was acceptable in comparison with 2.9% for a survey about precision medicine in South Korea [36], 16% for a pilot study for national cohort study [23] and 34% for a study about personalised health research in Switzerland [5]. We could not assess the extent of non-response bias in detail because only two characteristics were available for comparison; the proportion of responders according to household language, which was similar to that of the sample as a whole, and household size, with people from larger households less likely to respond than those from smaller households.

The proportion of survey participants reporting willingness to participate varies between studies. About 50% of potential participants in our study were willing to participate, which is comparable to findings from studies on personalised health research in Switzerland (54%) [5] and the United States (54%) [37], and clinical research in India (42%) [8]. Similarly, a large European survey involving 32 countries found that 53% of respondents were willing to provide personal information to a biobank [38]. In Australia, a telephone survey found that 61% of respondents were willing to participate in health research in general, with willingness increasing when specific information about the study was provided [39]. In Switzerland, 90% of respondents to an online survey reported willingness to participate in a national health cohort study [6]. The researchers suggested this high proportion could reflect a strong civic duty to support public research, particularly when led by the Federal Office of Public Health [6]. Indeed, altruism was one of the main reasons, stated by 56% of people who said that they would be willing to take part in a cohort study about pandemic preparedness. Willingness to take part in future research might also be overestimated, however, given that they selectively took part in the online survey. Actual participation may also be lower than intended participation. In studies in high-income countries, including ours, younger people, those with higher levels of education and/or higher incomes were more likely to say they would participate. Gender was not strongly associated with willingness to participate; the small number of respondents reporting a non-binary gender makes interpretation uncertain. We did not find any other studies about affective polarisation and willingness to participate in research.

## Interpretation of the study findings

Our investigation of affective polarisation as a dimension of willingness to participate in research adds to the knowledge in this field, particularly in the context of designing research that might investigate preventive interventions, such as vaccination against COVID-19. The finding that people who support vaccination were more willing to take part in a long-term cohort study than people who oppose vaccination is in line with general support for research by "good citizens" [3]. The

**Table 3. Effect modification by affective polarisation status of the association between opinion about COVID-19 vaccination and willingness to participate.**

| Affective polarisation status | Support vaccination | | Oppose vaccination | |
|---|---|---|---|---|
| | N² with outcome | aOR³ (95% CI) | N² with outcome | aOR³ (95% CI) |
| **Not polarised** | 97,556 | 1 | 56,783 | 0.53 (0.32 - 0.86) |
| **Polarised** | 216,219 | 1.51 (1.05 - 2.16) | 21,406 | 0.26 (0.12 - 0.56) |

¹Adjusted for age, education level, current work situation, income, household size, household location, gender, language spoken, foreign status, polarisation status, vaccine support status
²Weighted number of participants
³Marginal adjusted odds ratio

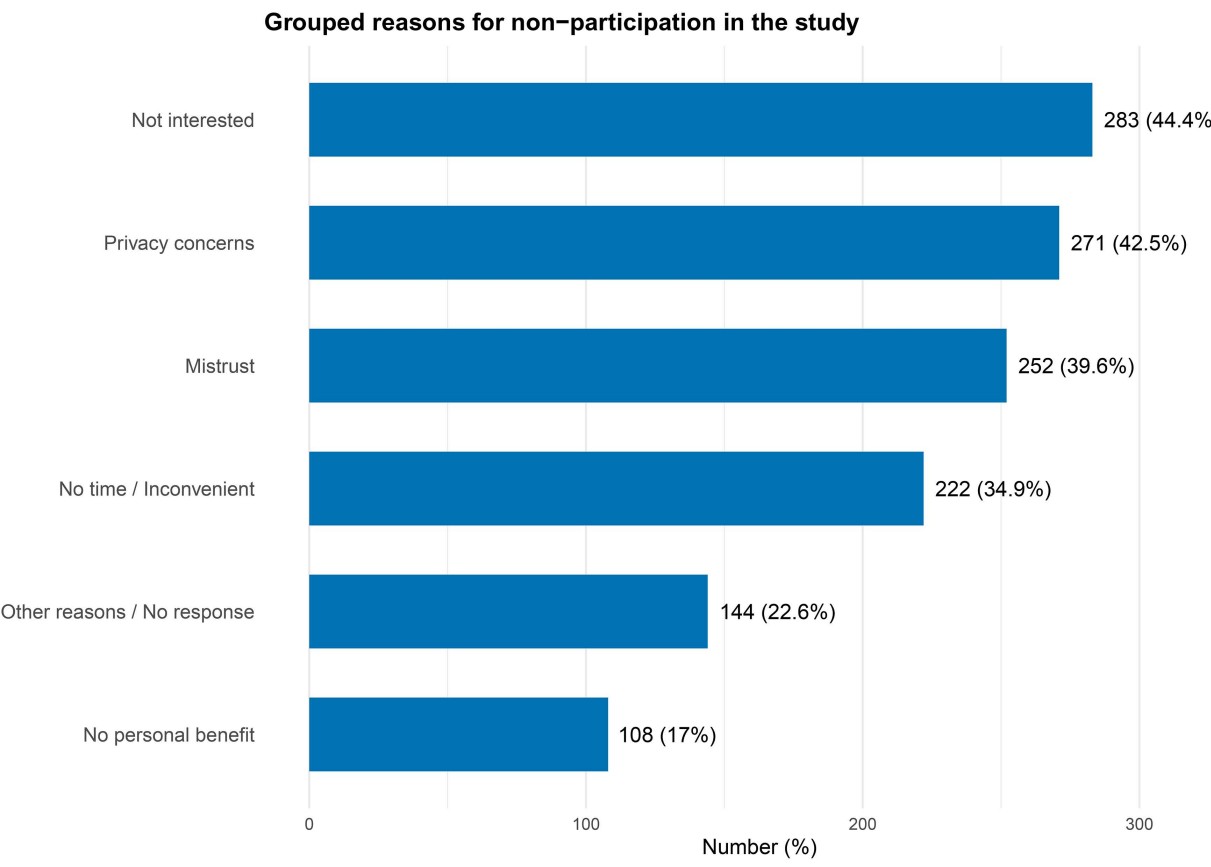

**Fig 2. Reasons for non-participation, amongst 637 respondents who stated that they would not be willing to take part in a cohort study.**

modifying effect of affective polarisation provides additional insight. People who supported COVID-19 vaccination were even more motivated to take part in research if they also held polarised views about other people's opinions. In contrast, both polarised and non-polarised opponents of vaccination reported lower willingness to participate than non-polarised supporters of vaccination. Existing research provides evidence of a link between affective polarisation and reduced vaccination uptake [40,41], lower concern about COVID-19, and decreased support for COVID-19 policies [42]. These findings align with our study if we consider participation in a long-term cohort study as a health-related behaviour. Additionally, we

found that mistrust and concerns about privacy of health data were among the top reasons for unwillingness to participate. We did not examine the association between affective polarisation and trust, but other researchers have found this in relation to vaccination and political opinions [43–45].

### Implications for research and public health

The findings from our study have implications for public health research and policy making. The strong associations between opinions about COVID-19 vaccination, vaccine-related affective polarisation and willingness to participate in future studies supports a multidisciplinary approach to research and its communication [9]. The relationship between affective polarisation and willingness to participate in public health research should also be extended to different contexts and countries with varying levels of affective polarisation [15] to examine the generalisability of our findings. Strategies to overcome under-representation of people who are older, less well educated, have lower incomes and have more negative opinions about preventive measures in research for pandemic preparedness are also needed to reduce the potential for participation bias. Community engagement activities could raise awareness of the importance of research about infectious diseases and pandemic preparedness and reduce mistrust about sharing health data. Technological advances, which reduce barriers to participation, such as lack of time, including the use of online surveys and mailing of self-collected biological samples might also be beneficial. Policy makers will benefit from future research about potential pandemic threats, despite the difficulties of achieving and sustaining high levels of participation [1]. Our study examined willingness to participate in research after the emergency phase of the COVID-19 pandemic, which might be affected by pandemic fatigue and shifting attitudes to trust. A study in Switzerland found that affective polarisation in relation to COVID-19 vaccination was less marked in 2023 than in 2022 [45]. Longer term study of affective polarisation in pandemic preparedness cohort studies, such as the BEready cohort, should be conducted to determine whether, in the next pandemic, early-stage solidarity in a crisis increases participation. Understanding factors affecting willingness to participate could help develop future strategies to sustain long-term research engagement across diverse populations.

### Supporting information

**S1 Fig. Distribution of raw data from variables which were dichotomised.**
(PDF)

**S2 Fig. Study participants flow chart.**
(PDF)

**S1 Table. STROBE checklist for cross-sectional studies.**
(PDF)

**S2 Table. Descriptive table of variables included in the analysis.**
(PDF)

**S3 Table. Weighting levels and represented households.**
(PDF)

**S4 Table. Responders of the survey by household size and language.**
(PDF)

**S5 Table. Willingness to participate in a long-term cohort study, unweighted denominators and proportions.**
(PDF)

**S6 Table. Sensitivity analysis.**
(PDF)

**S7 Table. Reasons to participate or decline participation in a long-term study, unweighted.**
(PDF)

## Acknowledgments

The authors would like to acknowledge Jean-Benoît Rossel for his contribution to data preparation, Christiane Pelzer for her support in data management, and Martin Samuel Wohlfender for his help with data visualisation.

## Author contributions

**Conceptualization** Aziz Mert Ipekci, Eva Maria Hodel, Maximilian Filsinger, Markus Freitag, Annika Frahsa, Gilles Wandeler, Nicola Low.

**Data curation:** Aziz Mert Ipekci, Selina Wegmüller.

**Formal analysis:** Aziz Mert Ipekci.

**Methodology:** Aziz Mert Ipekci, Eva Maria Hodel, Maximilian Filsinger, Nicola Low.

**Project administration:** Eva Maria Hodel.

**Supervision:** Nicola Low.

**Visualization:** Aziz Mert Ipekci.

**Writing – original draft:** Aziz Mert Ipekci, Eva Maria Hodel, Maximilian Filsinger, Nicola Low.

**Writing – review & editing:** Aziz Mert Ipekci, Eva Maria Hodel, Maximilian Filsinger, Selina Wegmüller, Simone Schuller, Markus Freitag, Annika Frahsa, Gilles Wandeler, Nicola Low.

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
