## [Decision Letter · Decision Letter 0]

20 Jan 2026

PONE-D-25-54952Who would take part in a pandemic preparedness cohort study? The role of vaccine-related affective polarisation: cross-sectional surveyPLOS One

Dear Dr. Low,

Thank you for submitting your manuscript to PLOS ONE. After careful consideration, we feel that it has merit but does not fully meet PLOS ONE’s publication criteria as it currently stands. Therefore, we invite you to submit a revised version of the manuscript that addresses the points raised during the review process.

We look forward to receiving your revised manuscript.

Kind regards,

Krzysztof Malarz, D.Sc., Ph.D., M.Sc.

Academic Editor

PLOS One

Journal Requirements:

6.  We note that Figure 1 in your submission contain [map/satellite] images which may be copyrighted. All PLOS content is published under the Creative Commons Attribution License (CC BY 4.0), which means that the manuscript, images, and Supporting Information files will be freely available online, and any third party is permitted to access, download, copy, distribute, and use these materials in any way, even commercially, with proper attribution. For these reasons, we cannot publish previously copyrighted maps or satellite images created using proprietary data, such as Google software (Google Maps, Street View, and Earth). For more information, see our copyright guidelines: http://journals.plos.org/plosone/s/licenses-and-copyright.

Reviewers' comments:

Reviewer's Responses to Questions

**Comments to the Author**

1. Is the manuscript technically sound, and do the data support the conclusions?

Reviewer #1: Yes

2. Has the statistical analysis been performed appropriately and rigorously? 

Reviewer #1: Yes

3. Have the authors made all data underlying the findings in their manuscript fully available?

The PLOS Data policy requires authors to make all data underlying the findings described in their manuscript fully available without restriction, with rare exception (please refer to the Data Availability Statement in the manuscript PDF file). The data should be provided as part of the manuscript or its supporting information, or deposited to a public repository. For example, in addition to summary statistics, the data points behind means, medians and variance measures should be available. If there are restrictions on publicly sharing data—e.g. participant privacy or use of data from a third party—those must be specified.requires authors to make all data underlying the findings described in their manuscript fully available without restriction, with rare exception (please refer to the Data Availability Statement in the manuscript PDF file). The data should be provided as part of the manuscript or its supporting information, or deposited to a public repository. For example, in addition to summary statistics, the data points behind means, medians and variance measures should be available. If there are restrictions on publicly sharing data—e.g. participant privacy or use of data from a third party—those must be specified.requires authors to make all data underlying the findings described in their manuscript fully available without restriction, with rare exception (please refer to the Data Availability Statement in the manuscript PDF file). The data should be provided as part of the manuscript or its supporting information, or deposited to a public repository. For example, in addition to summary statistics, the data points behind means, medians and variance measures should be available. If there are restrictions on publicly sharing data—e.g. participant privacy or use of data from a third party—those must be specified.requires authors to make all data underlying the findings described in their manuscript fully available without restriction, with rare exception (please refer to the Data Availability Statement in the manuscript PDF file). The data should be provided as part of the manuscript or its supporting information, or deposited to a public repository. For example, in addition to summary statistics, the data points behind means, medians and variance measures should be available. If there are restrictions on publicly sharing data—e.g. participant privacy or use of data from a third party—those must be specified.

Reviewer #1: Yes

4. Is the manuscript presented in an intelligible fashion and written in standard English?

Reviewer #1: Yes

5. Review Comments to the Author

Reviewer #1: A "literature review" section (immediately before the "Methods" section) should be included to strengthen the work by providing arguments in favor of the variables included in the quantitative analysis. In this regard, the "literature review" section should reference empirical studies on vaccine hesitancy, particularly for COVID-19. A summary table of these studies, categorized by variables influencing vaccine hesitancy and highlighting the most relevant aspects (country and years studied, methodology used, and results), would be very useful.

Likewise, the "comparison with other studies" section should be more comprehensive, including more studies, and a table comparing the results of this work with those of others would also be helpful.

Although the variables included in the quantitative analysis are correct and commonly included in this field of research, it should be discussed why affective polarization is included as a variable and not other political factors that empirical literature has shown to influence attitudes toward the Covid-19 pandemic, such as political ideology, political affiliation, and trust in the government and political institutions. Likewise, the exclusion of a highly likely relevant variable, identified by the authors, namely altruism, should be discussed.

6. PLOS authors have the option to publish the peer review history of their article (what does this mean?). If published, this will include your full peer review and any attached files.). If published, this will include your full peer review and any attached files.). If published, this will include your full peer review and any attached files.). If published, this will include your full peer review and any attached files.

...

Reviewer #1: No

---

## [Author Response · Author response to Decision Letter 1]

12 Mar 2026

We would like to thank the editor and the reviewer for their comments and for the opportunity to revise our manuscript.

We have numbered the editor’s and reviewer’s comments and copied them below. We respond to each comment and describe the changes made in the revised version of the manuscript. We give the line numbers for the location of the revised text in the clean version.

Editor’s comments

1) Authors' response: We have added the financial disclosure statement to the cover letter. We have resubmitted the figure files, following the guidelines.

2) If applicable, we recommend that you deposit your laboratory protocols in protocols.io to enhance the reproducibility of your results. Protocols.io assigns your protocol its own identifier (DOI) so that it can be cited independently in the future.

Authors' response: The request does not apply to our study, which did not use laboratory-based methods. Nevertheless, we aimed to ensure reproducibility through detailed reporting of the study design, data sources, and analytical methods in the Methods section.

3) Please ensure that your manuscript meets PLOS ONE's style requirements, including those for file naming. The PLOS ONE style templates can be found at https://journals.plos.org/plosone/s/file?id=wjVg/PLOSOne_formatting_sample_main_body.pdf and https://journals.plos.org/plosone/s/file?id=ba62/PLOSOne_formatting_sample_title_authors_affiliations.pdf

Authors' response: We would like to thank the editor for this reminder. We have followed the style templates for the text and supplementary material and ensured they are correctly referenced throughout the manuscript.

4) Your ethics statement should only appear in the Methods section of your manuscript. If your ethics statement is written in any section besides the Methods, please delete it from any other section.

Authors' response: We deleted the additional ethics statement at the end of the manuscript and retained the ethics statement only in the Methods section (page 4, lines 97-99).

5) Please provide a complete Data Availability Statement in the submission form, ensuring you include all necessary access information or a reason for why you are unable to make your data freely accessible.

Authors' response: We thank the editor for this request. We appreciate the benefits of publicly available datasets. Due to concerns about the potential for identifying individuals, our data cannot be shared publicly. Please find our Data Availability statement in our response to point 6, below.

6) We note that you have indicated that there are restrictions to data sharing for this study. For studies involving human research participant data or other sensitive data, we encourage authors to share de-identified or anonymized data. However, when data cannot be publicly shared for ethical reasons, we allow authors to make their data sets available upon request. For information on unacceptable data access restrictions, please see http://journals.plos.org/plosone/s/data-availability#loc-unacceptable-data-access-restrictions. If there are ethical or legal restrictions on sharing a de-identified data set, please explain them in detail (e.g., data contain potentially identifying or sensitive patient information, data are owned by a third-party organization, etc.) and who has imposed them (e.g., a Research Ethics Committee or Institutional Review Board, etc.). Please also provide contact information for a data access committee, ethics committee, or other institutional body to which data requests may be sent.

Authors' response: Our Data Availability statement has been updated in the submission form, and reads. “The data from this study are not publicly available because of the risk of identification of an individual participant. Some municipalities in the study area had very few respondents, and the characteristics reported could be combined to identify a person. Data are available upon reasonable request. Researchers interested in accessing the data from this online survey can request it by contacting the project management team at BEready.mcid@unibe.ch or by filling in the request form (https://www.beready.unibe.ch/for_researchers/). De-identified data will be made available to qualified researchers for approved analyses, and access will be granted following review and approval of a study proposal, establishment of a data use and transfer agreement, and successful ethical approval. Restrictions may apply depending on the specific nature of the data requested to ensure participant privacy and compliance with institutional policies. A shared dataset will exclude study participants from municipalities with fewer than 5 respondents. Fees may apply for data preparation and transfer, depending on the nature of the collaboration. We encourage multidisciplinary collaboration on projects that align with the goals of our study, particularly those focused on pandemic preparedness.”

7) We note that the grant information you provided in the ‘Funding Information’ and ‘Financial Disclosure’ sections do not match. When you resubmit, please ensure that you provide the correct grant numbers for the awards you received for your study in the ‘Funding Information’ section.

Authors' response: We confirm that we have written the same grant numbers in the ‘Funding Information’ and ‘Financial Disclosure’ sections. The wording of the Financial Disclosure (p18, lines 381-383) is, “This work was supported by the Multidisciplinary Center for Infectious Diseases, University of Bern (grant number MA21 awarded to AF, MarF, NL and the BEready project, no grant number, awarded to NL, GW).” Please note that this statement lists more than one co-author for each grant. The automated Funding Information section in the submission portal only allows one co-author to be associated with a grant, so this information only includes the first of the authors listed in the statement.

8) We note that Figure 1 in your submission contain [map/satellite] images which may be copyrighted. All PLOS content is published under the Creative Commons Attribution License (CC BY 4.0), which means that the manuscript, images, and Supporting Information files will be freely available online, and any third party is permitted to access, download, copy, distribute, and use these materials in any way, even commercially, with proper attribution.

Authors' response: We appreciate the editors for pointing this out. We have received confirmation from Swiss Federal Office of Public Health, in an email dated 25.02.2026, that, “There are no fees or special authorization for using standard services (Internet). It is important to cite the source (‘Federal Statistical Office, name of statistics + year’ <e.g. ‘Federal Statistical Office, Structural Survey 2017’>).”

Figure 1 was created using two datasets: swissBOUNDARIES3D and the official directory of towns and cities, both published by the Federal Office of Topography swisstopo (swisstopo.admin.ch), and the lakes and rivers 2025_GEOM_TK dataset of Swiss Federal Statistical Office (bfs.admin.ch).

We have added citations to the Federal Office of Topography and the Federal Statistical Office (ref 31, 32) and web addresses to the caption of Figure 1 (page 9, line 214-218).

9) Please include captions for your Supporting Information files at the end of your manuscript, and update any in-text citations to match accordingly. Please see our Supporting Information guidelines for more information: http://journals.plos.org/plosone/s/supporting-information.

Authors' response: We have included captions for our Supporting Information files at the end of our manuscript (page 27, lines 560-569). We have reorganised the supplementary materials to comply with the journal’s requirements and matched all in-text citations accordingly.

Authors' response: Thank you for this advice. The reviewer did not ask us to cite specific works. The reviewer did ask us to provide additional citations to support statements in the Introduction and Discussion, Comparison with other studies sections. We agreed with the comments and chose relevant references ourselves. In the Introduction, we added one reference (number 18). In the discussion, we added two references (numbers 38 and 39).

Authors' response: We have verified that our reference list contains no retracted articles. Our response to the reviewer includes all additional references, and the location in the text where they are cited.

Reviewer’s comments

1) A "literature review" section (immediately before the "Methods" section) should be included to strengthen the work by providing arguments in favor of the variables included in the quantitative analysis. In this regard, the "literature review" section should reference empirical studies on vaccine hesitancy, particularly for COVID-19. A summary table of these studies, categorized by variables influencing vaccine hesitancy and highlighting the most relevant aspects (country and years studied, methodology used, and results), would be very useful.

Authors' response: Thank you for the feedback about the need for a relevant literature review in the Introduction. We appreciate the opportunity to clarify the scope of our work. We apologise if we gave the reviewer the wrong impression about the subject of our study. To clarify, we have not used the words ‘vaccine hesitancy’ in any part of the manuscript. Our paper is about willingness to take part in research about pandemic preparedness – and how this might be affected by people’s feelings and opinions. We assessed feelings using a measure of affective polarisation in relation to views about COVID-19 vaccination – both positive and negative. Our review of the literature in the Introduction therefore included: studies about factors associated with willingness to participate in research before the COVID-19 pandemic (refs 4-6, cited p3); reasons why the COVID-19 pandemic might have affected willingness to participate (refs 7-11, cited p3); and the relevance of polarisation and affective polarisation (refs 12-17, cited p4). To address the reviewer’s comment, we made changes to:

- Abstract: we have made the subject of the study clearer at the start of the Abstract, “Willingness to participate in research is associated with multiple factors. Little is known about the role of people’s feelings, especially in the aftermath of the COVID-19 pandemic.” (page 2, lines 21-24).

- Introduction: we added a new reference, which is a systematic review about affective polarisation related to COVID-19 vaccination “Whilst affective polarisation was first defined and studied in the United States of America to understand [14], the COVID-19 pandemic revealed associations between affective polarisation and health behaviours such as COVID-19 vaccination uptake, adherence to pandemic countermeasures or anxiety about COVID-19 [14-18]” (page 3, line 80, ref 18).

We have followed the submission guidelines of the journal to provide “a brief review of the key literature.” Given that our literature review covers a range of different issues to put the study into context, we did not think it appropriate to tabulate these studies.

2) Likewise, the "comparison with other studies" section should be more comprehensive, including more studies, and a table comparing the results of this work with those of others would also be helpful.

Authors' response: We thank the reviewer for this constructive suggestion. We have expanded the "Comparison with other studies" section. We added two relevant studies (refs 38, 39), giving examples of other types of population-based health research (page 16, line 308-312), “Similarly, a large European survey involving 32 countries found that 53.3% of respondents were willing to provide personal information to a biobank [38]. In Australia, a telephone survey found that 61% of respondents were willing to participate in health research in general, with willingness increasing when specific information about the study was provided [39].” We believe that the six cited studies provide a concise but sufficient narrative overview of the current literature.

Regarding the suggested table, we would like to retain the description within the narrative of the text. These studies are examples of other types of research about willingness to participate. We feel that a table with detailed study-level information might give the impression of a comprehensive systematic literature search, which is beyond the scope of our study. We believe that introducing a table with a large amount of additional data at this stage might confuse the reader, potentially distracting from the primary focus of our study.

3) Although the variables included in the quantitative analysis are correct and commonly included in this field of research, it should be discussed why affective polarization is included as a variable and not other political factors that empirical literature has shown to influence attitudes toward the Covid-19 pandemic, such as political ideology, political affiliation, and trust in the government and political institutions.

Authors' response: We thank the reviewer for this comment and for the opportunity to articulate our choices more clearly. We agree that there is already empirical literature about political factors. We now acknowledge this explicitly in the Introduction (page 3-4, lines 72-74), “Political polarisation, according to ideology, partisan affiliation or trust in political institutions, has characterised responses to many aspects of COVID-19 prevention [12-13].”

We found a gap in the evidence about the role of affective polarisation, however. We have revised the introduction to make this point more clearly (page 4, lines 80-84), “Measures of people’s feelings about public health measures such as COVID-19 vaccination, and how they view others, are more proximate to health-associated behaviours than partisan or ideological differences. Whether affective polarisation is associated with willingness to take part in research about pandemics more generally is not well understood.”

In practice, the number of questions permitted in the questionnaire was restricted. We acknowledge this limitation in the Discussion, Strengths and limitations, including three new references (33, 34, 35) (page 16, lines 289-293), “Third, the number of questions in the survey was restricted since a lengthy questionnaire might have decreased the response rate [33]. Whilst there is evidence that political ideology and trust in government institutions are associated with an individual’s willingness to participate in scientific research [34, 35], we were not able to include questions about a broader range of political variables.”

4) Likewise, the exclusion of a highly likely relevant variable, identified by the authors, namely altruism, should be discussed.

Authors' response: We thank the reviewer for raising this important point. We agree that altruism is indeed a relevant factor in willingness to participate in research. Unfortunately, it was not measured as a standalone variable in our survey and therefore coul

---

## [Decision Letter · Decision Letter 1]

19 Mar 2026

Who would take part in a pandemic preparedness cohort study? The role of vaccine-related affective polarisation: cross-sectional survey

PONE-D-25-54952R1

Dear Dr. Low,

We’re pleased to inform you that your manuscript has been judged scientifically suitable for publication and will be formally accepted for publication once it meets all outstanding technical requirements.

Kind regards,

Angelo Moretti, Ph.D.

Academic Editor

PLOS One

Additional Editor Comments (optional):

Reviewers' comments:

Reviewer's Responses to Questions

**Comments to the Author**

1. If the authors have adequately addressed your comments raised in a previous round of review and you feel that this manuscript is now acceptable for publication, you may indicate that here to bypass the “Comments to the Author” section, enter your conflict of interest statement in the “Confidential to Editor” section, and submit your "Accept" recommendation.

Reviewer #1: All comments have been addressed

2. Is the manuscript technically sound, and do the data support the conclusions?

Reviewer #1: Yes

3. Has the statistical analysis been performed appropriately and rigorously? 

Reviewer #1: Yes

4. Have the authors made all data underlying the findings in their manuscript fully available?

The PLOS Data policy requires authors to make all data underlying the findings described in their manuscript fully available without restriction, with rare exception (please refer to the Data Availability Statement in the manuscript PDF file). The data should be provided as part of the manuscript or its supporting information, or deposited to a public repository. For example, in addition to summary statistics, the data points behind means, medians and variance measures should be available. If there are restrictions on publicly sharing data—e.g. participant privacy or use of data from a third party—those must be specified.requires authors to make all data underlying the findings described in their manuscript fully available without restriction, with rare exception (please refer to the Data Availability Statement in the manuscript PDF file). The data should be provided as part of the manuscript or its supporting information, or deposited to a public repository. For example, in addition to summary statistics, the data points behind means, medians and variance measures should be available. If there are restrictions on publicly sharing data—e.g. participant privacy or use of data from a third party—those must be specified.requires authors to make all data underlying the findings described in their manuscript fully available without restriction, with rare exception (please refer to the Data Availability Statement in the manuscript PDF file). The data should be provided as part of the manuscript or its supporting information, or deposited to a public repository. For example, in addition to summary statistics, the data points behind means, medians and variance measures should be available. If there are restrictions on publicly sharing data—e.g. participant privacy or use of data from a third party—those must be specified.requires authors to make all data underlying the findings described in their manuscript fully available without restriction, with rare exception (please refer to the Data Availability Statement in the manuscript PDF file). The data should be provided as part of the manuscript or its supporting information, or deposited to a public repository. For example, in addition to summary statistics, the data points behind means, medians and variance measures should be available. If there are restrictions on publicly sharing data—e.g. participant privacy or use of data from a third party—those must be specified.

Reviewer #1: Yes

5. Is the manuscript presented in an intelligible fashion and written in standard English?

Reviewer #1: Yes

6. Review Comments to the Author

Reviewer #1: (No Response)

7. PLOS authors have the option to publish the peer review history of their article (what does this mean?). If published, this will include your full peer review and any attached files.). If published, this will include your full peer review and any attached files.). If published, this will include your full peer review and any attached files.). If published, this will include your full peer review and any attached files.

...

Reviewer #1: **Yes:**Pedro Atienza MonteroPedro Atienza MonteroPedro Atienza MonteroPedro Atienza Montero

---

## [Editor Report · Acceptance letter]

PONE-D-25-54952R1

PLOS One

Dear Dr. Low,

I'm pleased to inform you that your manuscript has been deemed suitable for publication in PLOS One. Congratulations! Your manuscript is now being handed over to our production team.

Kind regards,

on behalf of

Dr. Angelo Moretti

Academic Editor

PLOS One